# Detection of SARS-CoV-2 in Saliva and Nasopharyngeal Swabs According to Viral Variants

Maud Salmona,[a,b] Marie-Laure Chaix,[a,c] Linda Feghoul,[a,b] Nadia Mahjoub,[a] Sarah Maylin,[a] Nathalie Schnepf,[a] Hervé Jacquier,[d,e] Eve-Marie Walle,[a] Marion Helary,[a] Guillaume Mellon,[f] Nathalie Osinski,[f] Widad Zebiche,[f] Yacine Achili,[g] Rishma Amarsy,[h] Véronique Mahé,[g] Jérôme Le Goff,[a,b] Constance Delaugerre[a,c]

[a]Laboratoire de Virologie, AP-HP, Hôpital Saint Louis, Paris, France
[b]Université Paris Cité, INSERM U976, Equipe INSIGHT, Paris, France
[c]Université Paris Cité, INSERM, Paris, France
[d]Service de Bactériologie-Hygiène, Hôpitaux Universitaires Saint-Louis - Lariboisière - Fernand Widal, Paris, France
[e]Université Paris Cité, IAME UMR 1137, INSERM, Paris, France
[f]Équipe Opérationnelle d'Hygiène, AP-HP, Hôpital Saint Louis, Paris, France
[g]Service Central de Santé au Travail, AP-HP, Hôpital Saint Louis, Paris, France
[h]Equipe Opérationnelle d'Hygiène, AP-HP, Hôpitaux Lariboisière-Fernand Widal, Paris, France

**ABSTRACT** The genome of the Omicron variant of concern (VOC) contains more than 50 mutations, many of which have been associated with increased transmissibility, differing disease severity, and the potential to elute immune responses acquired after severe acute respiratory syndrome coronavirus 2 (SARS-CoV-2) vaccination or infection with previous VOCs. Due to a better tropism for the upper respiratory tract, it was suggested that the detection of the Omicron variant could be preferred in saliva, compared to nasopharyngeal swabs (NPS). Our objective was to compare the SARS-CoV-2 levels in saliva fluid and NPS to estimated Ct values, according to the main SARS-CoV-2 variants circulating in France since the beginning of 2021. We analyzed 1,289 positive reverse transcription-polymerase chain reaction (RT-PCR) results during the three major waves: Alpha, Delta, and Omicron. NPS and saliva sampling were performed for 909 (71%) and 380 (29%) cases, respectively. The Ct values were significantly lower in the NPS samples than in the saliva samples for the three main VOCs. Still, the difference was less pronounced with the Omicron variant than for the Alpha and Delta variants. In contrast, in the saliva samples, Ct values were significantly lower for the Omicron variant than for the Delta (difference of $-2.7$ Ct) and the Alpha (difference of $-3.0$ Ct) variants, confirming a higher viral load in saliva. To conclude, the higher viral load in saliva was evidenced for the Omicron variant, compared to the Alpha and Delta variants, suggesting that established diagnostic methods might require revalidation with the emergence of novel variants.

**IMPORTANCE** Established methods for SARS-CoV-2 diagnostics might require revalidation with the emergence of novel variants. This is important for screening strategy programs and for the investigation of the characteristics of new variants in terms of tropism modification and increased viral burden leading to its spread. SARS-CoV-2 RT-PCR screening on saliva samples reported lower but acceptable performance, compared to nasopharyngeal samples. Due to a better tropism for the upper respiratory tract, it was suggested that the detection of the Omicron variant could be preferred in saliva, compared to nasopharyngeal swabs. Our study analyzed 1,289 positive RT-PCR results during the three major waves in France: Alpha, Delta, and Omicron. Our findings also showed a higher viral load in saliva for the Omicron variant, compared to the Alpha and Delta variants.

**Keywords** SARS-CoV-2, VOC, saliva, cycle threshold, Omicron

Address correspondence to Maud Salmona, maud.salmona@aphp.fr.

The authors declare no conflict of interest.

O n November 26, 2021, the World Health Organization (WHO) designated the newly detected B.1.1.529 lineage of severe acute respiratory syndrome coronavirus 2 (SARS-CoV-2), the Omicron variant of concern (VOC) (1). The genome of the Omicron VOC contains more than 50 mutations, many of which have been associated with increased transmissibility, differing disease severity, and the potential to evade immune responses acquired after SARS-CoV-2 vaccination or infection with previous VOCs (Alpha and Delta) (2). Interestingly Omicron's hyper-transmissibility was not associated with severe disease, as was observed with the Delta VOC.

Laboratory studies demonstrated that replication was similar for Omicron and Delta virus isolates in human nasal epithelial cultures. However, in lower airway organoids, lung cells, and gut cells, Omicron revealed lower replication (3). In a hamster model, omicron shows decreased lung infectivity and is less pathogenic than Delta and ancestral SARS-CoV-2 (4).

The detection of the Omicron genome via reverse transcription-polymerase chain reaction (RT-PCR) seems similar between saliva swabs and midturbinate swabs, whereas testing performance is less sensitive for the other VOCs (5–7).

Our objective was to compare SARS-CoV-2 levels in saliva fluid and nasopharyngeal swabs, as estimated by Ct values, according to the main SARS-CoV-2 variants that were circulating in France since the beginning of 2021: Alpha, Delta, and Omicron.

## RESULTS

During the study period (January 2021 to February 2022), 1,289 positive RT-PCR results were analyzed, corresponding to 1,244 health care workers (HCW) (Table 1). The median (interquartile range [IQR]) age was 36 years (28 to 47), and 933 (75%) of the individuals were women. NPS and saliva sampling were performed for 909 (71%) and 380 (29%) cases, respectively. The RT-PCR were done using different assays, such as the Cobas SARS-CoV-2 assay (ROC) ($n$ = 782, 61%), Alinity m SARS-CoV-2 assay (ALI) ($n$ = 249, 19%), and NeuMoDx SARS-CoV-2 assay (NMDx) ($n$ = 151, 12%). Overall, the median value of Ct was 26.7 (IQR: 21.1 to 31.5), ranging from 10.3 to 42.6. Variant determination was not performed or was uninformative for 760 (41%) samples, principally due to the Ct value being above 30. The SARS-CoV-2 variants analyzed were mainly Omicron ($n$ = 274, 52%), Alpha ($n$ = 140, 26%), and Delta ($n$ = 68, 13%). Among the 134 samples also sequenced via whole-genome sequencing (WGS), Nextclade (https://clades.nextstrain.org/) retrieved the following lineages: 29 Wuhan (no VOC), 19 Alpha (including 4 Alpha + 484K), 11 Beta, 33 Delta, and 42 Omicron (including 40 from the BA.1 lineage and 2 from the BA.2 lineage). These results were consistent with RT-PCR variant-specific screening test results for all samples. According to the date of sampling, RT-PCR results were available for the Omicron wave in 828 (64%), for the Alpha wave in 340 (26%), and the Delta wave in 121 (10%) positive cases.

Table 2 compares the characteristics, according to the type of sampling, NPS, or saliva. The median Ct value (IQR) was 24.4 (19.8 to 31.3) and 29.0 (25.8 to 31.7) in NPS and saliva, respectively ($P < 0.001$). NPS was performed using ROC in 483 (53%) cases, ALI in 175 (19%) cases, and NMDx in 147 (16%) cases, whereas saliva tests were performed using ROC in 299 (79%) cases and ALI in 74 (19%) cases. The distribution of sample type (NPS or saliva) was different, according to the different waves ($P < 0.05$). Fig. 1 showed the Ct distribution according to the sampling time for all of the 1,289 positive PCR results (Fig. 1A) and for those with variant determination ($n$ = 529) (Fig. 1B). Locally estimated scatterplot smoothing (LOESS) curves showed higher Ct values for saliva than for NPS. After January 2022, a convergence of the LOESS NPS and saliva curves was observed during the Omicron wave, due to lower Ct values for saliva samples (Fig. 1A and B).

The comparison of Ct values according to the main 3 VOC showed a significantly lower median Ct value in the NPS samples compared to the saliva samples: Alpha (19.8 [16.5 to 26.5], $n$ = 108 versus 28.2 [26.9 to 30.7], $n$ = 32; $P < 0.001$), Delta (20.3 [17.5 to 26.5], $n$ = 48 versus 27.9 [26.2 to 29.2], $n$ = 20; $P < 0.001$), and Omicron (22.4 [19.7 to 26.0], $n$ =202 versus 25.2 [23.2 to 27.8], $n$ = 72; $P < 0.001$) (Fig. S1).

We also compared the median Ct values between Alpha, Delta, and Omicron for each type of sample (NPS and saliva) (Fig. 2). For the NPS samples, the Omicron Ct

**TABLE 1** Characteristics of sampling

| Sampling characteristic | Number of samples |
|---|---|
| Number of patients | 1,244 |
| | |
| Gender, *n* (%) | |
| Female | 933 (75%) |
| Male | 311 (25%) |
| | |
| Age, median (IQR) yrs | 36 (28 to 47) |
| | |
| Number of samples | 1,289 |
| NPS, *n* (%) | 909 (71%) |
| Saliva, *n* (%) | 380 (29%) |
| | |
| RT-PCR assays | |
| ROC | 782 (61%) |
| ALI | 249 (19%) |
| NMDx | 151 (12%) |
| CPH | 28 (2.2%) |
| SPX | 79 (6.1%) |
| | |
| Cycle threshold, Ct, median (IQR) | 27 (21, 31) |
| | |
| Variant of concern, VOC, *n* (%) | |
| Available | 529 |
| Alpha | 140 (26%) |
| Beta/Gamma | 13 (2.5%) |
| Number of patients | 1,244 |
| Omicron | 274 (52%) |
| Other | 34 (6.4%) |
| Not available (NA) | 760 |
| | |
| Period of sampling, *n* (%) | |
| Period 1, Alpha wave | 340 (26%) |
| Period 2, Delta wave | 121 (10%) |
| Period 3, Omicron wave | 828 (64%) |

values were significantly higher than those of the Alpha variant ($P = 0.0018$) but were not significantly different from those of the Delta variant ($P = 0.052$) (Fig. 2A). Interestingly, in the saliva samples, Omicron Ct values were significantly lower than those of Alpha ($P < 0.001$) and Delta ($P = 0.0033$), respectively (Fig. 2B). To exclude any bias introduced by the different RT-PCR tests assays used, the analysis was restricted to the ROC test and confirmed the lower Ct values in the saliva samples with Omicron compared to those obtained with Alpha or Delta (Fig. S2A and B).

## DISCUSSION

The reference standard for SARS-CoV-2 screening is RT-PCR on nasopharyngeal samples. Screening on saliva samples displayed lower but acceptable performances (5, 6). This screening strategy was implemented in cases of repeated testing, particularly in HCW, and in facilitating mass screening in schools or hospitals. Due to a better tropism for the upper respiratory tract, it was suggested that the detection of the Omicron variant could be preferred in saliva, compared to NPS (7). Our study analyzed 1,289 positive RT-PCR results during the three major waves in France: Alpha, Delta, and Omicron. Globally SARS-CoV-2 RT-PCR Ct values were significantly lower (corresponding to higher viral loads) in NPS samples than in saliva samples. Still, the difference was less pronounced with the Omicron variant than for the Alpha and Delta variants. Between NPS and saliva, a difference of 3.0 Ct (corresponding roughly to 1 log of copies/mL) was observed for Omicron, a difference of 7.6 Ct (2.5 log) for Delta and a difference of 8.4 Ct (2.8 log) for Alpha. Interestingly, when we compared the Ct values in the NPS samples, Omicron Ct values were similar to the Delta Ct values and higher than

**TABLE 2** Comparison between NPS and saliva samples

| Sampling characteristic | NPS | Saliva | P value |
|---|---|---|---|
| N | 909 | 380 | |
| Ct, median (IQR) | 24 (20, 31) | 29 (26, 32) | <0.001 |
| VOC | | | 0.5 |
| Available | 396 | 133 | |
| Alpha | 108 (27%) | 32 (24%) | |
| Beta/Gamma | 12 (3%) | 1 (0.8%) | |
| Delta | 48 (12%) | 20 (15%) | |
| Omicron | 202 (51%) | 72 (54%) | |
| Other | 26 (6.6%) | 8 (6%) | |
| Not available (NA) | 513 | 247 | |
| RT-PCR assays | | | <0.001 |
| ROC | 483 (53%) | 299 (79%) | |
| ALI | 175 (19%) | 74 (19%) | |
| NMDx | 147 (16%) | 4 (1.1%) | |
| CPH | 25 (2.8%) | 3 (0.8%) | |
| SPX | 79 (8.7%) | 0 | |
| Period of sampling, n (%) | | | 0.05 |
| Period 1, Alpha wave | 279 (31%) | 61 (16%) | |
| Period 2, Delta wave | 73 (8%) | 48 (13%) | |
| Period 3, Omicron wave | 557 (61%) | 271 (71%) | |

the Alpha Ct values. In contrast, in the saliva samples, the Ct values were significantly lower for the Omicron variant than for the Delta (difference of $-2.7$ Ct) and Alpha (difference of $-3.0$ Ct) variants, confirming a higher viral load in saliva.

Recently, Marais et al. described that the Omicron variant might be more readily detected using RT-PCR in saliva swabs of 382 symptomatic patients, compared to paired mild-turbinate swabs (7). Another study did not support a preferred sample type for Omicron detection but suggested the heterogeneous distribution of viral loads in mild-turbinate, oropharyngeal swabs, and in saliva sample types collected from 121 symptomatic persons (8). These studies were conducted on a limited number of positive cases, did not use the same RT-PCR assay for the saliva and nasal samples (8), and did not compare viral loads between different variants. A strength of our study was that it confirmed that the viral loads in saliva were higher for the Omicron variant, even when it was restricted to a unique RT-PCR assay that was previously validated for saliva samples (5).

Our study presents some limitations. First, information on symptoms and vaccination was not collected, and saliva sample types could be used more often for systematic and iterative screening in asymptomatic HCW. However, this would probably not affect our conclusions, as lower viral loads were mostly described in persons without symptoms, compared with symptomatic individuals (9). Second, the primary analysis was performed with different RT-PCR assays, but the second analysis was restricted to the ROC assay, representing 60% of the tests, and confirmed the results. Third, we did not use NPS and saliva paired samples for a strict comparison but did report an analysis per period covering the three major waves in France of the Alpha, Delta, and Omicron variants. In a previous study (10), Chu et al. reported the improvement of the performance of saliva versus NPS mild-turbinate sampling before and after the replacement with the Beta VOC.

To conclude, established diagnostic methods might require revalidation with the emergence of novel variants. This is important for screening strategy programs and the investigation of the characteristics of new variants in terms of tropism modification and increased viral burden leading to its spread.

## MATERIALS AND METHODS

**Participants.** For testing for SARS-CoV-2 infections, nasopharyngeal swab (NPS) and saliva sampling were offered from the beginning of 2021 to the health care workers (HCW) of Saint Louis Hospital, Paris, France.

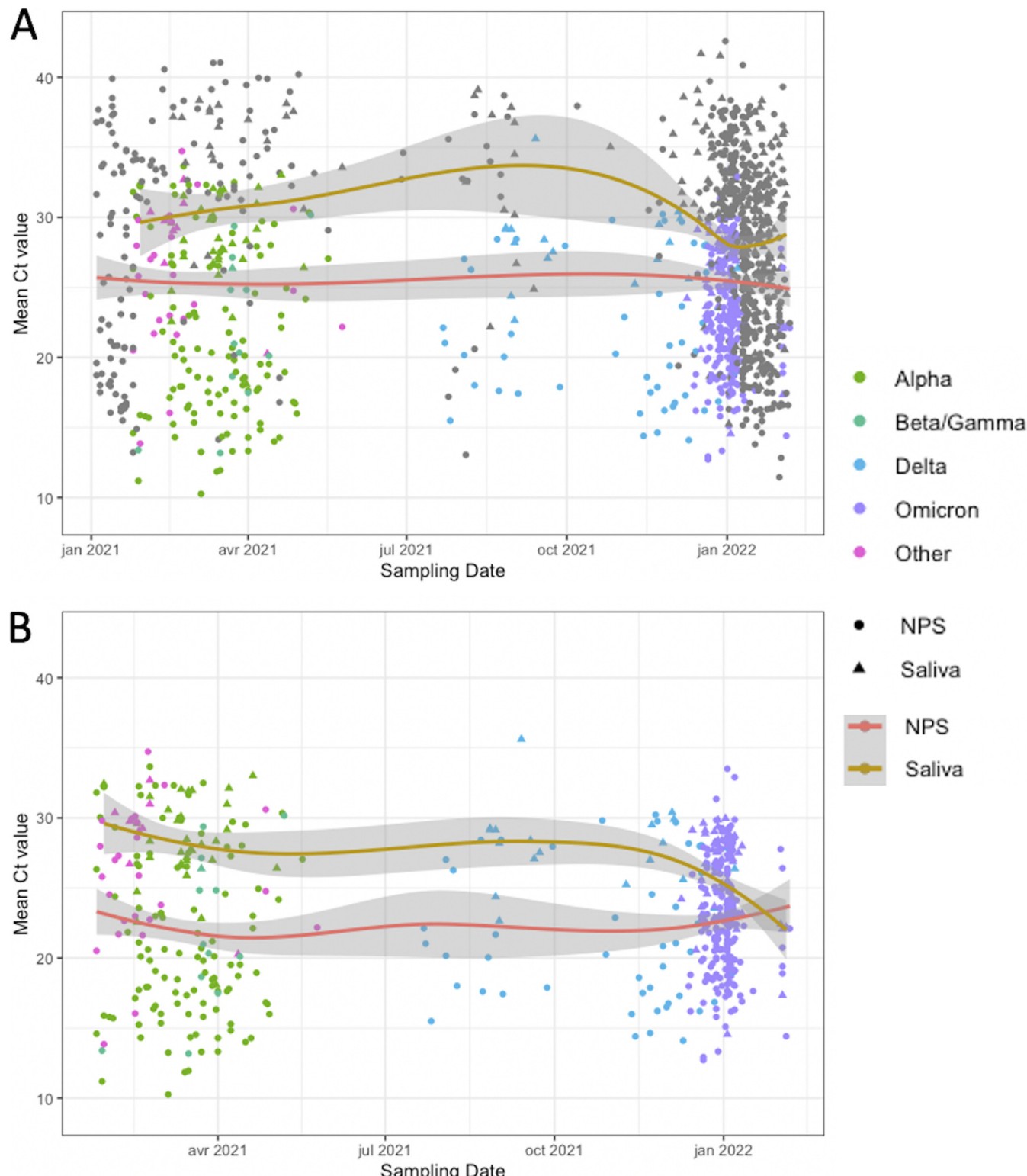

**FIG 1** SARS-CoV-2 Cycle Threshold (Ct) values as a function of time and sampling type. (A) For all of the 1,289 positive PCR results. (B) For the 529 positive PCR results with a result of the variant determination. The data points represent SARS-CoV-2-positive samples. The shapes represent the sample type, and the color represents the variant determination. Grey points reflect ungenotyped samples. LOESS curves are shown for nasopharyngeal swab samples (red) and saliva samples (orange). The shaded areas represent 95% confidence intervals.

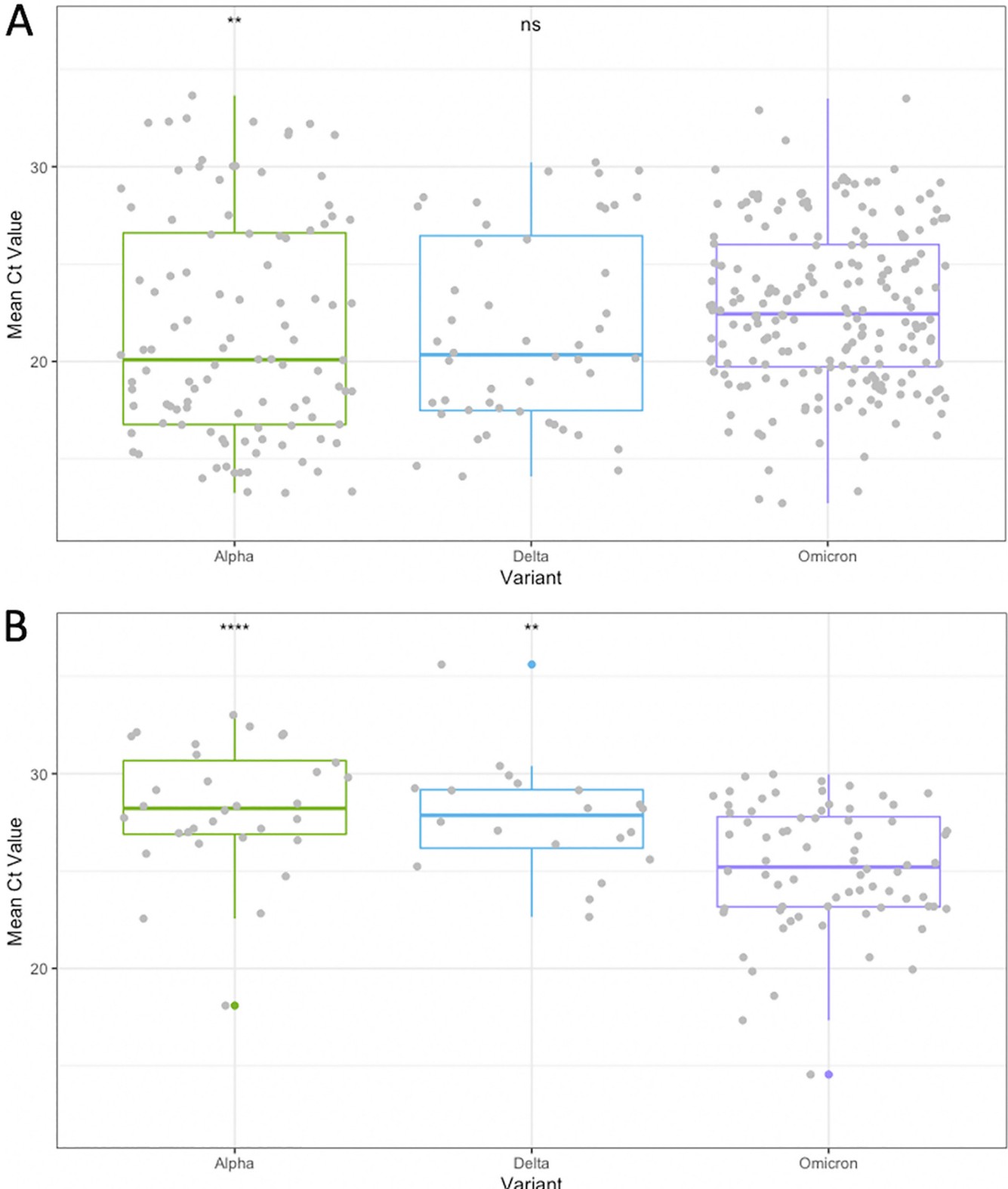

**FIG 2** SARS-CoV-2 Ct values among the Alpha, Delta, and Omicron VOCs in nasopharyngeal swab (NPS) (A) and saliva (B) samples. Boxes range from the first to third quartiles. Midlines represent median values. Individual points represent SARS-CoV-2-positive samples. Statistically significant differences (*, $P < 0.05$; **, $P < 0.01$; ***, $P < 0.001$) of Alpha and Delta versus Omicron were determined by either a Mann-Whitney U-test or a Student's $t$ test.

The main circumstances were COVID-19 symptoms, contact tracing, return from vacation, and iterative screening, but these pieces of information were not collected at the time of sampling.

**PCR testing.** NPS were collected by a health care professional and were placed directly onto a viral transport medium before transport to the laboratory. Fresh, self-collected saliva were transported to the laboratory without any viral transport media and were then diluted in phosphate-buffered saline (PBS) and lysis buffer before amplification (5).

Molecular detection was based on the RT-PCR amplification of at least two viral genome regions. The assays used by our laboratory during the study period were the Cobas SARS-CoV-2 assay ("ROC", Roche Molecular Diagnostics, Mannheim, Germany), Alinity m SARS-CoV-2 assay ("ALI", Abbott Molecular, Rungis, France), NeuMoDx SARS-CoV-2 assay ("NMDx", Qiagen, Courtaboeuf, France), Xpert Xpress SARS-CoV-2 assay ("CPH", Cepheid, Maurens-Scopont, France), and Simplexa COVID-19 Direct assay ("SPX", DiaSorin, Antony, France).

Our laboratory information system collected results expressed as cycle threshold (Ct) values. For assays with multiple targets, the mean Ct value was calculated.

**Variant determination.** RT-PCR variant-specific screening tests were performed for positive samples with Ct < 30. Screening tests analyzed the presence of different mutations in the SARS-CoV-2 spike protein. The following deletion and mutations were analyzed during the Alpha wave: Δ69-70, N501Y, and E484K. The Alpha VOC was assigned with the presence of both Δ69-70 and N501. The Beta/Gamma VOC was set for a profile with both N501Y and E484K mutations without Δ69-70. The following mutations were analyzed during the Delta and Omicron waves: E484K, E484Q, L452R, and K417N. The presence of L452R without E484K/Q was assigned to the Delta VOC. The absence of L452R associated with the presence of K417N corresponded to the Omicron VOC. Due to the virological distribution of the SARS-CoV-2 strains at the time of data analysis, virological results were expressed as Alpha, Beta/Gamma, Omicron, Delta, Other, or NA ("Not Available"). The designation of NA was used when screening was impossible or gave no interpretable result (low viral load).

Whole-genome sequencing was performed for 134 samples with RT-PCR variant-specific screening test results. Briefly, a Qiaseq SARS-Cov-2 Prime Panel (Qiagen, Hilden, Germany) was used to create a tiled amplicon across the SARS-Cov-2 genome. DNA libraries were prepared using Nextera XT and were sequenced using an Illumina MiSeq with a 300 v2 cartridge (Illumina, San Diego, CA USA).

**Statistical analyses.** For positive samples, the RT-PCR Ct value and variant information were collected for each participant at screening for SARS-CoV-2 infection. In the case of reinfection, defined as a positive RT-PCR result 60 days after the first infection, two RT-PCR Ct results were retained in the analysis. All of the available RT-PCR results were analyzed between January 1, 2021, and February 6, 2022. The primary analysis described the Ct values for NPS and saliva sampling as Alpha, Beta/Gamma, Omicron, Delta, Other, or NA (combined in "Other"). Then, the analysis was restricted to the main three circulating VOCs: Alpha, Delta, and Omicron.

For the supplementary analyses, we defined three periods, according to the circulation of the predominant VOC in France, as declared by the national survey of Santé Publique France (https://www.santepubliquefrance.fr/). Period 1 was the "Alpha wave" from January 1st, 2021, to June 22, 2021. Period 2 was the "Delta wave" from June 23, 2021 to December 19, 2021. Period 3 was the "Omicron wave" from December 20, 2021 to February 6, 2022. Then, the analysis was done using the RT-PCR results provided by the ROC assay to exclude any technical bias.

The descriptive and statistical analyses were performed with R, using the ggplot2, ggpubr, and gt summary packages with a significance level of $P < 0.05$.

**Ethical statement.** The study was conducted in accordance with the Declaration of Helsinki. This study was a noninterventional study with no additional sampling beyond the usual procedures. Biological material and clinical data were obtained only for standard viral diagnostics, following physicians' prescriptions (no specific sampling, no modification of the sampling protocol). Data analyses were conducted using an anonymized database. According to the French Health Public Law (CSP Art L 1121-1.1), such protocol are exempt from informed consent applications.

**Data availability.** The SARS-Cov-2 whole-genome sequences can be found in the GISAID database (https://gisaid.org/). GISAID virus name numbers are detailed in the supplemental material.

## SUPPLEMENTAL MATERIAL

Supplemental material is available online only.
**SUPPLEMENTAL FILE 1**, PDF file, 0.7 MB.

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
