## [Reviewer comments · Microbiology Spectrum]

Microbiology Spectrum

Detection of SARS-COV-2 in saliva and naso-pharyngeal swab according to viral variants

Maud Salmona, Marie-Laure Chaix, Linda Feghoul, Nadia Mahjoub, Sarah MAYLIN, Nathalie Schnepf, Hervé Jacquier, Eve-Marie Walle, Marion Helary, Guillaume Mellon, Nathalie Osinski, Widad Zebiche, Yacine Achili, Rishma Amarsy, Veronique Mahé, Jérôme Le Goff, and Constance Delaugerre

Corresponding Author(s): Maud Salmona, APHP Hospital Saint Louis

Review Timeline:

Submission Date:	June 7, 2022
Editorial Decision:	August 12, 2022
Revision Received:	September 13, 2022
Editorial Decision:	September 19, 2022
Revision Received:	September 20, 2022
Editorial Decision:	October 12, 2022
Revision Received:	October 13, 2022
Accepted:	October 19, 2022

Editor: Daniel Perez

Reviewer(s): Disclosure of reviewer identity is with reference to reviewer comments included in decision letter(s). The following individuals involved in review of your submission have agreed to reveal their identity: Diana Hardie (Reviewer #1)

Transaction Report:

DOI: <https://doi.org/10.1128/spectrum.02133-22>

August 12, 2022

Dr. Maud Salmona
APHP Hospital Saint Louis
1 av Claude Vellefaux
Paris 75010
France

Re: Spectrum02133-22 (Detection of SARS-COV-2 in saliva and naso-pharyngeal swab according to viral variants)

Dear Dr. Maud Salmona:

Thank you for submitting your manuscript to Microbiology Spectrum. Please note that in order to speed up the review process and in an attempt to avoid further delays, I am relying on the review of single reviewer and my own assessment. When submitting the revised version of your paper, please provide (1) point-by-point responses to the issues raised by the reviewers as file type "Response to Reviewers," not in your cover letter, and (2) a PDF file that indicates the changes from the original submission (by highlighting or underlining the changes) as file type "Marked Up Manuscript - For Review Only". Please use this link to submit your revised manuscript - we strongly recommend that you submit your paper within the next 60 days or reach out to me. Detailed instructions on submitting your revised paper are below.

Link Not Available

Sincerely,

Daniel Perez

Journals Department
Reviewer comments:

Reviewer #1 (Comments for the Author):

This study evaluated the SARS-CoV-2 PCR CT values in diagnostic nasopharyngeal or saliva samples over a period of 3 infection waves in France in 2021/2. Each wave was caused by a VOC (alpha, delta and omicron). The aim was to determine whether variant type was associated with a change in the relative levels of virus present in saliva and NP samples (as reflected in the CT value). Variant calling was based partly on timing of sample collection (relative to defined waves) and also on rapid PCR typing using particular variant specific SNPs or deletions. Loess curves of CT data were used to show mean trends in positive NP and saliva samples in the different epidemic waves.

The mean CT value in positive NP samples was consistently lower than for saliva in all waves. However, the CT difference

between NP and saliva was less marked in omicron infections at 3.0 vs 7.6 for delta and 8.4 for alpha, this is in keeping with a higher level of shedding in this compartment with omicron infection, relative to previous variants.

The study provides further data on the altered shedding kinetics associated with omicron infection.

The paper is well written and clear to follow.

1. My main issue is with the variant typing protocol used to confirm the omicron variant. Omicron variant was defined on the basis of 2 SNPs in the spike gene, namely K417N and wt L452. While the timing of the wave makes these almost certainly omicron, some more evidence would be helpful to verify that these were indeed omicron (this pattern could also be beta). Was WGS done on any of these to verify the lineage assignment? Lines 105-107 in the manuscript mention that whole genome sequencing was performed on 134 samples with 100% concordance, but then give a reference (8) which is not correct. I would like to know more detail on the verification done on the 134 samples.
2. Delta assignment: lines 99-101 state that L452R and E484Q was used to define delta. To my knowledge E484Q is a kappa specific mutation. Delta has wild type E484 at this position. Please confirm and rewrite this section.
3. Line 56: evade, not elute
4. Line 139: "...performed for 909 (71%) and 380 (29%) cases, respectively. The RT-PCR were done..."

Staff Comments:

Preparing Revision Guidelines

Please return the manuscript within 60 days; if you cannot complete the modification within this time period, please contact me. If you do not wish to modify the manuscript and prefer to submit it to another journal, please notify me of your decision immediately so that the manuscript may be formally withdrawn from consideration by Microbiology Spectrum.

Response to Reviewers

Reviewer #1 (Comments for the Author): This study evaluated the SARS-CoV-2 PCR CT values in diagnostic nasopharyngeal or saliva samples over a period of 3 infection waves in France in 2021/2. Each wave was caused by a VOC (alpha, delta and omicron). The aim was to determine whether variant type was associated with a change in the relative levels of virus present in saliva and NP samples (as reflected in the CT value). Variant calling was based partly on timing of sample collection (relative to defined waves) and also on rapid PCR typing using particular variant specific SNPs or deletions. Loess curves of CT data were used to show mean trends in positive NP and saliva samples in the different epidemic waves. The mean CT value in positive NP samples was consistently lower than for saliva in all waves. However, the CT difference between NP and saliva was less marked in omicron infections at 3.0 vs 7.6 for delta and 8.4 for alpha, this is in keeping with a higher level of shedding in this compartment with omicron infection, relative to previous variants. The study provides further data on the altered shedding kinetics associated with omicron infection. The paper is well written and clear to follow.

1. My main issue is with the variant typing protocol used to confirm the omicron variant. Omicron variant was defined on the basis of 2 SNPs in the spike gene, namely K417N and wt L452. While the timing of the wave makes these almost certainly omicron, some more evidence would be helpful to verify that these were indeed omicron (this pattern could also be beta). Was WGS done on any of these to verify the lineage assignment? Lines 105-107 in the manuscript mention that whole genome sequencing was performed on 134 samples with 100% concordance, but then give a reference (8) which is not correct. I would like to know more detail on the verification done on the the 134 samples.

Thanks to the reviewer to highlight this issue. Among the 134 samples analyzed in WGS, we retrieved the following lineages: 29 wuhan, 19 Alpha (including 4 Alpha+484K), 11 Beta, 33 Delta and 42 Omicrons (21J)). These results were consistent with the screening results for all samples (the 29 Wuhan were determined to be "Other" by the PCR screening). We are sorry for the reference which was indeed not correct, we have removed it from the manuscript. We have modified the paragraph in the following way : « *Whole-genome sequencing was performed for 134 samples with RT-PCR variant-specific screening test results. Briefly, Qiaseq SARS-Cov-2 Prime Panel (Qiagen, Hilden, Germany) was used to create tiled amplicon across the SARS-Cov-2 genome. DNA libraries were prepared using Nextera XT and sequenced using Illumina Miseq with a 300 v2 cartridge (Illumina, San Diego, CA USA). Among the 134 samples sequenced with WGS, Nextclade (<https://clades.nextstrain.org/>) retrieved the following lineage: 29 Wuhan (no VOC), 19 Alpha (including 4 Alpha+484K), 11 Beta, 33 Delta and 42 Omicron VOCs. These results were consistent with RT-PCR variant-specific screening test results for all samples.* »

2. Delta assignment: lines 99-101 state that L452R and E484Q was used to define delta. To my knowledge E484Q is a kappa specific mutation. Delta has wild type E484 at this position. Please confirm and rewrite this section.

We are sorry, our sentence was not clear. Indeed, it is the absence of E484K/Q and the presence of 452R which allowed to define Delta VOC. To clarify, we have modified the sentence in the following way: " The presence of L452R without E484K/Q was assigned to Delta VOC" (line 99/100)

3. Line 56: evade, not elute

4. Line 139: "...performed for 909 (71%) and 380 (29%) cases, respectively. The RT-PCR were done..."

We have made the changes requested by the reviewer.

September 19, 2022

Dr. Maud Salmona
APHP Hospital Saint Louis
1 av Claude Vellefaux
Paris 75010
France

Re: Spectrum02133-22R1 (Detection of SARS-COV-2 in saliva and naso-pharyngeal swab according to viral variants)

Dear Dr. Maud Salmona:

Thank you for submitting your manuscript to Microbiology Spectrum. Please address the minor comments of the reviewer and send back. The ms will be considered accepted after those minor changes are addressed. When submitting the revised version of your paper, please provide (1) point-by-point responses to the issues raised by the reviewers as file type "Response to Reviewers," not in your cover letter, and (2) a PDF file that indicates the changes from the original submission (by highlighting or underlining the changes) as file type "Marked Up Manuscript - For Review Only". Please use this link to submit your revised manuscript - we strongly recommend that you submit your paper within the next 60 days or reach out to me. Detailed instructions on submitting your revised paper are below.

Link Not Available

Sincerely,

Daniel Perez

Journals Department
Reviewer comments:

Reviewer #1 (Comments for the Author):

. I have the following minor issues to resolve:

1. Lines 121-124: move to results. For the 42 omicron samples that underwent whole genome sequencing, what was the predominant sub genotype? i.e indicate whether these were BA.1 or 2 (or other).
2. Figure 1 (a and b): add to the colour code key that grey dot/triangle reflects un-genotyped samples.

Staff Comments:

Preparing Revision Guidelines

Please return the manuscript within 60 days; if you cannot complete the modification within this time period, please contact me. If you do not wish to modify the manuscript and prefer to submit it to another journal, please notify me of your decision immediately so that the manuscript may be formally withdrawn from consideration by Microbiology Spectrum.

Spectrum review August 2022

This study evaluated the SARS-CoV-2 PCR CT values in diagnostic nasopharyngeal or saliva samples over a period of 3 infection waves in France in 2021/2. Each wave was caused by a VOC (alpha, delta and omicron). The aim was to determine whether variant type was associated with a change in the relative levels of virus present in saliva and NP samples (as reflected in the CT value). Variant calling was based partly on timing of sample collection (relative to defined waves) and also on rapid PCR typing using particular variant specific SNPs or deletions. Loess curves of CT data were used to show mean trends in positive NP and saliva samples in the different epidemic waves.

The mean CT value in positive NP samples was consistently lower than for saliva in all waves. However, the CT difference between NP and saliva was less marked in omicron infections at 3.0 vs 7.6 for delta and 8.4 for alpha, this is in keeping with a higher level of shedding in this compartment with omicron infection, relative to previous variants.

The study provides further data on the altered shedding kinetics associated with omicron infection.

The paper is well written and clear to follow.

1. My main issue is with the variant typing protocol used to confirm the omicron variant. Omicron variant was defined on the basis of 2 SNPs in the spike gene, namely K417N and wt L452. While the timing of the wave makes these almost certainly omicron, some more evidence would be helpful to verify that these were indeed omicron (this pattern could also be beta). Was WGS done on any of these to verify the lineage assignment? Lines 105-107 in the manuscript mention that whole genome sequencing was performed on 134 samples with 100% concordance, but then give a reference (8) which is not correct. I would like to know more detail on the verification done on the 134 samples.
2. Delta assignment: lines 99-101 state that L452R and E484Q was used to define delta. To my knowledge E484Q is a kappa specific mutation. Delta has wild type E484 at this position. Please confirm and rewrite this section.
3. Line 56: evade, not elute
4. Line 139: "...performed for 909 (71%) and 380 (29%) cases, respectively. The RT-PCR were done..."

Review of the revised manuscript: 19 September 2022

I am satisfied that the points raised in the original review have been addressed. I have the following minor issues to resolve:

1. Lines 121-124: move to results. For the 42 omicron samples that underwent whole genome sequencing, what was the predominant sub genotype? i.e indicate whether these were BA.1 or 2 (or other).
2. Figure 1 (a and b): add to the colour code key that grey dot/triangle reflects un-genotyped samples.

Response to Reviewers

Reviewer #1 (Comments for the Author):

I have the following minor issues to resolve:

1. Lines 121-124: move to results. For the 42 omicron samples that underwent whole genome sequencing, what was the predominant sub genotype? i.e indicate whether these were BA.1 or 2 (or other).

We have moved the paragraph to the results section as requested by the reviewer.

Concerning the 42 Omicrons, 40 were BA.1 and 2 were BA.2, we have added this information in the paragraph.

2. Figure 1 (a and b): add to the colour code key that grey dot/triangle reflects un-genotyped samples

We have made the changes requested by the reviewer.

October 12, 2022

Dr. Maud Salmona
APHP Hospital Saint Louis
1 av Claude Vellefaux
Paris 75010
France

Re: Spectrum02133-22R2 (Detection of SARS-COV-2 in saliva and naso-pharyngeal swab according to viral variants)

Dear Dr. Maud Salmona:

Thank you for submitting your manuscript to Microbiology Spectrum. I noticed that the ms describes whole genome sequencing. Accession numbers for the 134 sequences must be provided as well as a DATA AVAILABILITY statement indicating those numbers and database where they have been deposited. When submitting the revised version of your paper, please provide (1) point-by-point responses to the issues raised by the reviewers as file type "Response to Reviewers," not in your cover letter, and (2) a PDF file that indicates the changes from the original submission (by highlighting or underlining the changes) as file type "Marked Up Manuscript - For Review Only". Please use this link to submit your revised manuscript - we strongly recommend that you submit your paper within the next 60 days or reach out to me. Detailed instructions on submitting your revised paper are below.

Link Not Available

Sincerely,

Daniel Perez

Journals Department
Reviewer comments:

Staff Comments:

Preparing Revision Guidelines

To submit your modified manuscript, log onto the eJP submission site at <https://spectrum.msubmit.net/cgi-bin/main.plex>. Go to Author Tasks and click the appropriate manuscript title to begin the revision process. The information that you entered when you first submitted the paper will be displayed. Please update the information as necessary. Here are a few examples of required

updates that authors must address:

Please return the manuscript within 60 days; if you cannot complete the modification within this time period, please contact me. If you do not wish to modify the manuscript and prefer to submit it to another journal, please notify me of your decision immediately so that the manuscript may be formally withdrawn from consideration by Microbiology Spectrum.

Response to Reviewers

Reviewer #1 (Comments for the Author):

Thank you for submitting your manuscript to Microbiology Spectrum. I noticed that the ms describes whole genome sequencing. Accession numbers for the 134 sequences must be provided as well as a DATA AVAILABILITY statement indicating those numbers and database where they have been deposited.

We add the following paragraph lines 148- 150

“Data Availability Statement:

SARS-Cov-2 whole genome sequences can be found in the GISAID database (<https://gisaid.org/>) , GISAID Virus name numbers are detailed in the supplementary material.”

We also add GISAID Virus name numbers in the Supplementary Mat.

hCov-19/France/IDF-SLS-762111069335/2021 ; hCov-19/France/IDF-SLS-762111068761/2021 ;
hCov-19/France/IDF-SLS-762111073787/2021 ; hCov-19/France/IDF-SLS-762111077079/2021 ;
hCov-19/France/IDF-SLS-762111083705/2021 ; hCov-19/France/IDF-SLS-762111080793/2021 ;
hCov-19/France/IDF-SLS-762111085044/2021 ; hCov-19/France/IDF-SLS-762111084651/2021 ;
hCov-19/France/IDF-SLS-762111091673/2021 ; hCov-19/France/IDF-SLS-762111093692/2021 ;
hCov-19/France/IDF-SLS-762111092796/2021 ; hCov-19/France/IDF-SLS-762112003325/2021 ;
hCov-19/France/IDF-SLS-762112008470/2021 ; hCov-19/France/IDF-SLS-762112009743/2021 ;
hCov-19/France/IDF-SLS-762112019105/2021 ; hCov-19/France/IDF-SLS-762112018531/2021 ;
hCov-19/France/IDF-SLS-762112018241/2021 ; hCov-19/France/IDF-SLS-762112028345/2021 ;
hCov-19/France/IDF-SLS-762112030150/2021 ; hCov-19/France/IDF-SLS-762112034673/2021 ;
hCov-19/France/IDF-SLS-762112044785/2021 ; hCov-19/France/IDF-SLS-762112042179/2021 ;
hCov-19/France/IDF-SLS-762112047939/2021 ; hCov-19/France/IDF-SLS-762112047101/2021 ;
hCov-19/France/IDF-SLS-762112051058/2021 ; hCov-19/France/IDF-SLS-762112052614/2021 ;
hCov-19/France/IDF-SLS-762112058059/2021 ; hCov-19/France/IDF-SLS-762112061178/2021 ;
hCov-19/France/IDF-SLS-762112061084/2021 ; hCov-19/France/IDF-SLS-762112064050/2021 ;
hCov-19/France/IDF-SLS-762112065420/2021 ; hCov-19/France/IDF-SLS-762112065423/2021 ;
hCov-19/France/IDF-SLS-762112071071/2021 ; hCov-19/France/IDF-SLS-762112066878/2021 ;
hCov-19/France/IDF-SLS-762112067406/2021 ; hCov-19/France/IDF-SLS-762112067482/2021 ;
hCov-19/France/IDF-SLS-762112068811/2021 ; hCov-19/France/IDF-SLS-762112068868/2021 ;
hCov-19/France/IDF-SLS-762112068912/2021 ; hCov-19/France/IDF-SLS-762112069011/2021 ;
hCov-19/France/IDF-SLS-762112069650/2021 ; hCov-19/France/IDF-SLS-762112071127/2021 ;
hCov-19/France/IDF-SLS-762112071771/2021 ; hCov-19/France/IDF-SLS-762112072654/2021 ;
hCov-19/France/IDF-SLS-762112073611/2021 ; hCov-19/France/IDF-SLS-762112076896/2021 ;
hCov-19/France/IDF-SLS-762112075551/2021 ; hCov-19/France/IDF-SLS-762112076117/2021 ;
hCov-19/France/IDF-SLS-762112076127/2021 ; hCov-19/France/IDF-SLS-762112076189/2021 ;
hCov-19/France/IDF-SLS-762112076940/2021 ; hCov-19/France/IDF-SLS-762112077426/2021 ;
hCov-19/France/IDF-SLS-762112079789/2021 ; hCov-19/France/IDF-SLS-762112079805/2021 ;
hCov-19/France/IDF-SLS-762112079816/2021 ; hCov-19/France/IDF-SLS-762112089662/2021 ;
hCov-19/France/IDF-SLS-762112090173/2021 ; hCov-19/France/IDF-SLS-762112089531/2021 ;
hCov-19/France/IDF-SLS-762112089650/2021 ; hCov-19/France/IDF-SLS-762112089699/2021 ;
hCov-19/France/IDF-SLS-762112090264/2021 ; hCov-19/France/IDF-SLS-762112090514/2021 ;
hCov-19/France/IDF-SLS-762112090527/2021 ; hCov-19/France/IDF-SLS-762112090974/2021 ;
hCov-19/France/IDF-SLS-762201003960/2022 ; hCov-19/France/IDF-SLS-762201003652/2022 ;
hCov-19/France/IDF-SLS-762201030716/2022 ; hCov-19/France/IDF-SLS-762201030963/2022 ;

hCov-19/France/IDF-SLS-762201031748/2022 ; hCov-19/France/IDF-SLS-762202003824/2022 ;
hCov-19/France/IDF-SLS-762202007355/2022 ; hCov-19/France/IDF-SLS-762101101664/2021 ;
hCov-19/France/IDF-SLS-762101101751/2021 ; hCov-19/France/IDF-SLS-762102003663/2021 ;
hCov-19/France/IDF-SLS-762102003101/2021 ; hCov-19/France/IDF-SLS-762102011623/2021 ;
hCov-19/France/IDF-SLS-762102017734/2021 ; hCov-19/France/IDF-SLS-762102017876/2021 ;
hCov-19/France/IDF-SLS-762102015848/2021 ; hCov-19/France/IDF-SLS-762102018004/2021 ;
hCov-19/France/IDF-SLS-762102023608/2021 ; hCov-19/France/IDF-SLS-762102024194/2021 ;
hCov-19/France/IDF-SLS-762102029131/2021 ; hCov-19/France/IDF-SLS-762102032207/2021 ;
hCov-19/France/IDF-SLS-762102037054/2021 ; hCov-19/France/IDF-SLS-762102041403/2021 ;
hCov-19/France/IDF-SLS-762102047074/2021 ; hCov-19/France/IDF-SLS-762102048430/2021 ;
hCov-19/France/IDF-SLS-762102048996/2021 ; hCov-19/France/IDF-SLS-762102052308/2021 ;
hCov-19/France/IDF-SLS-762102055793/2021 ; hCov-19/France/IDF-SLS-762102058821/2021 ;
hCov-19/France/IDF-SLS-762102053035/2021 ; hCov-19/France/IDF-SLS-762102056673/2021 ;
hCov-19/France/IDF-SLS-762102059157/2021 ; hCov-19/France/IDF-SLS-762102058833/2021 ;
hCov-19/France/IDF-SLS-762102063781/2021 ; hCov-19/France/IDF-SLS-762102061345/2021 ;
hCov-19/France/IDF-SLS-762102070212/2021 ; hCov-19/France/IDF-SLS-762102072023/2021 ;
hCov-19/France/IDF-SLS-762102077510/2021 ; hCov-19/France/IDF-SLS-762102082352/2021 ;
hCov-19/France/IDF-SLS-762102083256/2021 ; hCov-19/France/IDF-SLS-762102083264/2021 ;
hCov-19/France/IDF-SLS-762103003119/2021 ; hCov-19/France/IDF-SLS-762103013192/2021 ;
hCov-19/France/IDF-SLS-762103059603/2021 ; hCov-19/France/IDF-SLS-762103080753/2021 ;
hCov-19/France/IDF-SLS-762103085636/2021 ; hCov-19/France/IDF-SLS-762103088014/2021 ;
hCov-19/France/IDF-SLS-762103090904/2021 ; hCov-19/France/IDF-SLS-762103091529/2021 ;
hCov-19/France/IDF-SLS-762103101811/2021 ; hCov-19/France/IDF-SLS-762104002994/2021 ;
hCov-19/France/IDF-SLS-762104038729/2021 ; hCov-19/France/IDF-SLS-762104043278/2021 ;
hCov-19/France/IDF-SLS-762104052661/2021 ; hCov-19/France/IDF-SLS-762104064987/2021 ;
hCov-19/France/IDF-SLS-762104097047/2021 ; hCov-19/France/IDF-SLS-762104098612/2021 ;
hCov-19/France/IDF-SLS-762105010666/2021 ; hCov-19/France/IDF-SLS-762105024259/2021 ;
hCov-19/France/IDF-SLS-762105080592/2021 ; hCov-19/France/IDF-SLS-762107057879/2021 ;
hCov-19/France/IDF-SLS-762107072141/2021 ; hCov-19/France/IDF-SLS-762108070774/2021 ;
hCov-19/France/IDF-SLS-762110087396/2021 ; hCov-19/France/IDF-SLS-762110097890/2021 ;
hCov-19/France/IDF-SLS-762111009230/2021 ; hCov-19/France/IDF-SLS-762111027078/2021 ;
hCov-19/France/IDF-SLS-762111036734/2021 ; hCov-19/France/IDF-SLS-762111040555/2021 ;
hCov-19/France/IDF-SLS-762111048861/2021 ; hCov-19/France/IDF-SLS-762111067035/2021

October 19, 2022

Dr. Maud Salmona
APHP Hospital Saint Louis
1 av Claude Vellefaux
Paris 75010
France

Re: Spectrum02133-22R3 (Detection of SARS-COV-2 in saliva and naso-pharyngeal swab according to viral variants)

Dear Dr. Maud Salmona:

Your manuscript has been accepted, and I am forwarding it to the ASM Journals Department for publication. You will be notified when your proofs are ready to be viewed.

Sincerely,

Daniel Perez
Editor, Microbiology Spectrum
